# Drift doesn't Matter: Dynamic Decomposition with Diffusion Reconstruction for Unstable Multivariate Time Series Anomaly Detection

**Chengsen Wang**[*]   **Zirui Zhuang**[*]   **Qi Qi**[†]   **Jingyu Wang**[†]
**Xingyu Wang**   **Haifeng Sun**   **Jianxin Liao**

State Key Laboratory of Networking and Switching Technology,
Beijing University of Posts and Telecommunications, Beijing, China
`{cswang, zhuangzirui, qiqi8266, wangjingyu}@bupt.edu.cn`
`{wangxingyu, hfsun, liaojx}@bupt.edu.cn`

## Abstract

Many unsupervised methods have recently been proposed for multivariate time series anomaly detection. However, existing works mainly focus on stable data yet often omit the drift generated from non-stationary environments, which may lead to numerous false alarms. We propose **D**ynamic **D**ecomposition with **D**iffusion **R**econstruction ($D^3R$), a novel anomaly detection network for real-world unstable data to fill the gap. $D^3R$ tackles the drift via decomposition and reconstruction. In the decomposition procedure, we utilize data-time mix-attention to dynamically decompose long-period multivariate time series, overcoming the limitation of the local sliding window. The information bottleneck is critical yet difficult to determine in the reconstruction procedure. To avoid retraining once the bottleneck changes, we control it externally by noise diffusion and directly reconstruct the polluted data. The whole model can be trained end-to-end. Extensive experiments on various real-world datasets demonstrate that $D^3R$ significantly outperforms existing methods, with a 11% average relative improvement over the previous SOTA models. Code is available at https://github.com/ForestsKing/D3R.

## 1 Introduction

Due to the rarity of anomalies, unsupervised anomaly detection of multivariate time series is an essential area in data mining and industrial applications. Reconstruction-based models are commonly used in unsupervised anomaly detection. The reconstruction error is small for normal series while large for abnormal series. Based on this principle, most anomalies can be detected without the label. In the real world, the temporal patterns typically change over time as they are generated from non-stationary environments. For example, the growth in the popularity of a service would cause customer metrics (e.g., request count) to drift upwards over time. Ignoring these factors would cause a deterioration in the performance of the anomaly detector. Despite the enormous advancements of previous research, most focus on stable data. Figure 1 illustrates how the anomaly scores provided by existing approaches tend to increase in the red area when the data is unstable, leading to false alarms.

To handle the unstable data, we attempt to decompose it into a stable component and a trend component, focusing more on the stable part in the reconstruction procedure. However, the trend part cannot be ignored completely, as the drift may be anomalous. There are still two challenges in this

---

[*]Equal contribution.
[†]Corresponding author.

37th Conference on Neural Information Processing Systems (NeurIPS 2023).

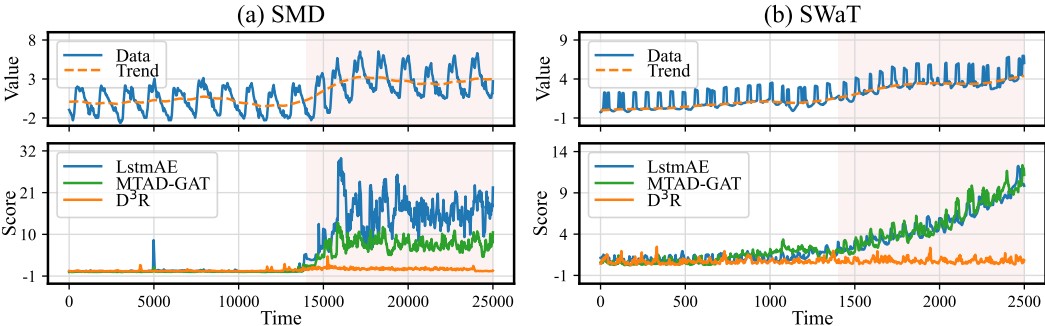

Figure 1: Anomaly scores of existing methods on SMD and SWaT datasets. All moments are normal. Vertical drift occurs in the red area. The moment with a higher anomaly score is more likely to be identified as an anomaly.

context: **Challenge 1**: Limitation of the decomposition for long-period time series. Most classical decomposition algorithms [2, 17, 27] are static, making it hard to be applied in the real world, where the data is updated in real time. With deep learning, some dynamic algorithms have been proposed; however, they are mainly based on averaging [29] or Fourier Transform [28, 26] within a local sliding window. Fundamentally, they can not apply to the data whose period is larger than the size of the sliding window. **Challenge 2**: High training cost of adjusting the information bottleneck in the reconstruction procedure. The information bottleneck is critical for reconstruction-based models, and adjusting it is difficult. If it is too small, normal data will be poorly reconstructed; if it is too large, abnormal data will be successfully reconstructed. Previous methods most rely on an internal information bottleneck, such as the latent space size. As the information bottleneck is an attribute of the model itself, any change to the bottleneck necessitates retraining the model.

After analyzing numerous data series, we systematically organize the distribution shift into two categories. Vertical drift refers to statistical characteristics such as mean changing over time, while horizontal drift refers to values shifting left or right at similar times in various periods. In response to the above challenges, we propose **D**ynamic **D**ecomposition with **D**iffusion **R**econstruction ($D^3R$) for long-period unstable multivariate time series anomaly detection. To solve Challenge 1, we utilize timestamps as external information to overcome the limitations of the local sliding window. Specifically, the data-time mix-attention and offset subtraction are utilized to solve vertical and horizontal drift, respectively. Moreover, we introduced a disturbance strategy in training to increase the robustness of the model. To solve Challenge 2, we propose a new method named noise diffusion to control the information bottleneck externally. The diffusion [6] provides a new view of the information bottleneck, treating the noise as a bottleneck and the unpolluted information as a condition. Because the bottleneck is no longer an attribute of the model itself, the different sizes can be set during revision without retraining the model.

The contribution of our paper is summarised as follows:

- A novel dynamic decomposition method for long-period multivariate time series is proposed. It effectively utilizes external information to overcome the limitations of the local sliding window.

- A new approach to control information bottleneck externally by noise diffusion is also proposed. It avoids the high training cost of adjusting the information bottleneck.

- Based on the findings for the unstable data when taking unsupervised anomaly detection, we propose a novel anomaly detection network called $D^3R$. The $D^3R$ achieves the new SOTA results on various real-world datasets and significantly outperforms baselines on unstable datasets.

## 2    Related work

As a significant real-world problem, unsupervised anomaly detection of multivariate time series has received much attention. According to the criterion for anomaly detection, the paradigms roughly fall

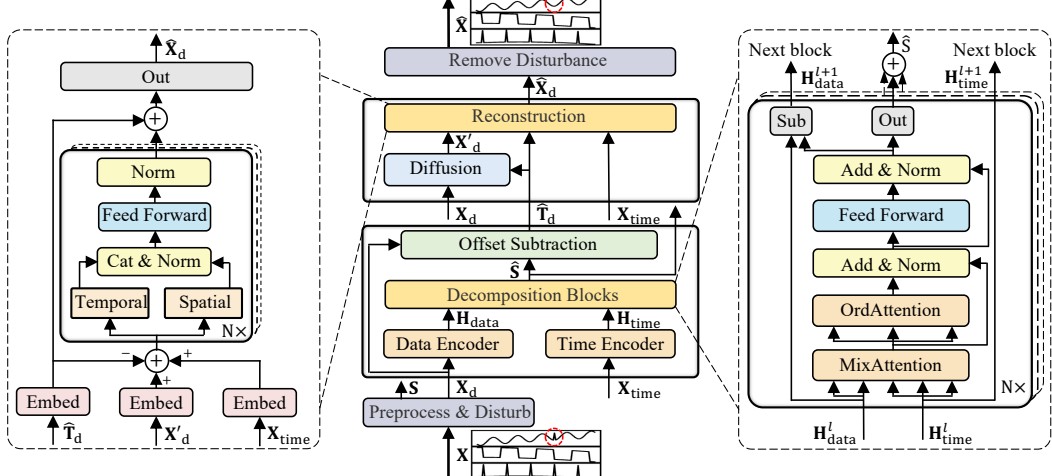

Figure 2: The architecture of D³R mainly consists of two modules: dynamic decomposition and diffusion reconstruction. The detailed architecture of the decomposition blocks is shown on the right panel. The detailed architecture of the reconstruction backbone network is shown on the left panel.

into the categories of probability-based, linear transformation-based, proximity-based, outlier-based, and neural network-based approaches.

As for the probability-based approach, [12, 13] are modeled by a statistical probability function on a multivariate cumulative distribution, providing anomaly scores based on the probability of occurrence of the sample. In the linear transformation-based method, [20, 22] begin by mapping the multivariate data before determining the boundary from the mapping space. Proximity-based algorithms, such as [18, 5], seek to cluster data based on similarity and then calculate intra-cluster and inter-cluster distances. In the outlier-based approach, [14, 16] compare the outliers degree of the testing samples with the training samples to identify the anomaly.

Most of the approaches mentioned above do not consider the temporal continuity of series. In recent years, neural network-based methods have become increasingly significant. Current neural network-based approaches can be divided into prediction-based and reconstruction-based. Although prediction-based methods [21, 8] are effective in modeling for the next timestamp, they are susceptible to interference from historical information. Reconstruction-based approaches [32, 11] do a great job capturing the distribution across the whole series. However, they are sensitive to the size of the information bottleneck.

Furthermore, existing works mainly focus on stable data, and their performance may suffer significantly in the real world, where the drift is frequent.

## 3 Method

An input of multivariate time series anomaly detection is denoted by $\mathbf{X} \in \mathbb{R}^{n \times k}$, where $n$ is the length of timestamps, and $k$ is the number of variables. The task is to produce an output vector $\mathbf{y} \in \mathbb{R}^n$, where $y_i \in \{0, 1\}$ denotes whether the $i^{th}$ timestamp is an anomaly.

### 3.1 Overview

The overall architecture of D³R is shown in Figure 2. The dynamic decomposition module first models the data and timestamp features by the data encoder and time encoder. Next, it uses stacked decomposition blocks to extract the stable component. Finally, we get the trend component by offset subtraction. The diffusion reconstruction module utilizes noise diffusion to construct information bottleneck externally and then directly reconstructs the polluted data by the backbone network. Reconstruction error is the anomaly score. In order to model both temporal dependency and dimension dependency, the data encoder and the reconstruction backbone network both consist of stacked spatial-

temporal transformer blocks. To increase the robustness of the model, we also propose a disturbance strategy during training.

## 3.2 Data preprocessing

We perform timestamp hard embedding, labeled stable component construction, and disturbance strategy for the input. Timestamp hard embedding is applied to the training and testing sets, while labeled stable component construction and disturbance strategy are applied to the training set only.

**Timestamp hard embedding**   To make better use of timestamps in anomaly detection, we hard-code the timestamps of $\mathbb{R}^{n \times 1}$ into an embedding $\mathbf{X}_{\text{time}} \in \mathbb{R}^{n \times 5}$ like [34, 29], with each dimension representing minute of the hour, hour of the day, day of the week, day of the month, and month of the year.

**Labeled stable component construction**   We extract the trend $\mathbf{T} \in \mathbb{R}^{n \times k}$ by moving average, then the labeled stable component $\mathbf{S} = \mathbf{X} - \mathbf{T}, \mathbf{S} \in \mathbb{R}^{n \times k}$ is obtained. The construction of the labeled stable components prevents our model from being disturbed when the training data is unstable.

**Disturbance strategy**   To increase the robustness of the model, we add a vertical drift $\mathbf{d} \in \mathbb{R}^k$ sampled from a $[-p, p]$ uniform distribution to each variable of the training data. The final input of dynamic decomposition module is $\mathbf{X}_{\text{d}} = \mathbf{X} + \mathbf{d}, \mathbf{X}_{\text{d}} \in \mathbb{R}^{n \times k}$.

## 3.3 Dynamic decomposition

The dynamic decomposition module consists of a data encoder, a time encoder, stacked decomposition blocks, and an offset subtraction. The data encoder is implemented based on the spatial-temporal transformer block, which captures the temporal and dimension dependency. The output of data encoder is $\mathbf{H}_{\text{data}} \in \mathbb{R}^{n \times d_{\text{model}}}$, where $d_{\text{model}}$ is hidden state dimension in the model. The time encoder consists only of the temporal transformer block, which models the temporal correlation of timestamps to obtain $\mathbf{H}_{\text{time}} \in \mathbb{R}^{n \times d_{\text{model}}}$. Data-time mix-attention constitutes the subject of stacked decomposition blocks, which are used to extract the stable component $\hat{\mathbf{S}} \in \mathbb{R}^{n \times k}$. Finally, to solve the challenges of horizontal drift, we obtain the trend component $\hat{\mathbf{T}}_{\text{d}} \in \mathbb{R}^{n \times k}$ through offset subtraction.

**Spatial-temporal transformer block**   The architecture of the spatial-temporal transformer block is shown in the solid line box on the left panel of Figure 2. Assuming the input of $l^{th}$ layer is $\mathbf{H}^l \in \mathbb{R}^{n \times d_{\text{model}}}$ with $n$ timestamps and $d_{\text{model}}$ dimensions. In the temporal transformer, we obtain temporal relationship $\mathbf{H}^l_{\text{temporal}} \in \mathbb{R}^{n \times d_{\text{model}}}$ by directly applying multi-head self-attention (MSA) to $n$ vectors of size $d_{\text{model}}$:

$$
\begin{aligned}
\hat{\mathbf{H}}^l_{\text{temporal}} &= \text{LayerNorm}\left(\mathbf{H}^l + \text{MSA}\left(\mathbf{H}^l, \mathbf{H}^l, \mathbf{H}^l\right)\right) \\
\mathbf{H}^l_{\text{temporal}} &= \text{LayerNorm}\left(\hat{\mathbf{H}}^l_{\text{temporal}} + \text{FeedForward}\left(\hat{\mathbf{H}}^l_{\text{temporal}}\right)\right)
\end{aligned}
\tag{1}
$$

where $\hat{\mathbf{H}}^l_{\text{temporal}}$ is intermediate variable. LayerNorm$(\cdot)$ denotes layer normalization as widely adopted in [25, 34, 29], FeedForward$(\cdot)$ denotes a multi-layer feedforward network, MSA$(\mathbf{Q}, \mathbf{K}, \mathbf{V})$ denotes the multi-head self-attention [25] layer where $\mathbf{Q}, \mathbf{K}, \mathbf{V}$ serve as queries, keys and values. In the spatial transformer, we obtain dimension relationship $\mathbf{H}^l_{\text{spatial}} \in \mathbb{R}^{n \times d_{\text{model}}}$ by applying MSA to $d_{\text{model}}$ vectors of size $n$:

$$
\begin{aligned}
\hat{\mathbf{H}}^l_{\text{spatial}} &= \text{LayerNorm}\left(\left(\mathbf{H}^l\right)^{\text{T}} + \text{MSA}\left(\left(\mathbf{H}^l\right)^{\text{T}}, \left(\mathbf{H}^l\right)^{\text{T}}, \left(\mathbf{H}^l\right)^{\text{T}}\right)\right) \\
\mathbf{H}^l_{\text{spatial}} &= \text{LayerNorm}\left(\left(\hat{\mathbf{H}}^l_{\text{spatial}}\right)^{\text{T}} + \text{FeedForward}\left(\left(\hat{\mathbf{H}}^l_{\text{spatial}}\right)^{\text{T}}\right)\right)
\end{aligned}
\tag{2}
$$

where $\hat{\mathbf{H}}^l_{\text{spatial}}$ is intermediate variable. $(\cdot)^{\text{T}}$ denotes the transposition of matrices. Finally, we obtain the input of $l + 1^{th}$ layer $\mathbf{H}^{l+1} \in \mathbb{R}^{n \times d_{\text{model}}}$ by:

$$
\begin{aligned}
\hat{\mathbf{H}}^{l+1} &= \text{LayerNorm}\left(\mathbf{H}^l_{\text{temporal}} \oplus \mathbf{H}^l_{\text{spatial}}\right) \\
\mathbf{H}^{l+1} &= \text{LayerNorm}\left(\text{FeedForward}\left(\hat{\mathbf{H}}^{l+1}\right)\right)
\end{aligned}
\tag{3}
$$

where $\widehat{\mathbf{H}}^{l+1} \in \mathbb{R}^{n \times 2d_{\text{model}}}$ is intermediate variable, and $\oplus$ represents concatenation.

**Data-time mix-attention**   The original self-attention $\mathcal{A}_{\text{o}}(\mathbf{Q}, \mathbf{K}, \mathbf{V})$ [25] only models the data information and ignores the role of timestamps. We define the data-time mix-attention as:

$$\mathcal{A}_{\text{m}}\left(\mathbf{Q}_{\text{d}}, \mathbf{K}_{\text{d}}, \mathbf{Q}_{\text{t}}, \mathbf{K}_{\text{t}}, \mathbf{V}_{\text{t}}\right) = \text{Softmax}\left(\frac{\mathbf{Q}_{\text{d}}\mathbf{K}_{\text{d}}^{\text{T}} + \mathbf{Q}_{\text{t}}\mathbf{K}_{\text{t}}^{\text{T}}}{\sqrt{d_{\text{k}}}}\right)\mathbf{V}_{\text{t}} \tag{4}$$

where $\text{Softmax}(\cdot)$ is conducted row by row like [25, 34, 29], $\mathbf{Q}_{\text{d}}, \mathbf{K}_{\text{d}} \in \mathbb{R}^{n \times d_{\text{k}}}$ are length-n data queries, data keys of $d_{\text{k}}$ dimension mapped from $\mathbf{H}_{\text{data}}$, respectively. $\mathbf{Q}_{\text{t}}, \mathbf{K}_{\text{t}}, \mathbf{V}_{\text{t}} \in \mathbb{R}^{n \times d_{\text{k}}}$ are time queries, time keys and time values of $d_{\text{k}}$ dimension mapped from $\mathbf{H}_{\text{time}}$, respectively. D³R learns to autonomously merge data and time information by mapping them into attention space.

**Decomposition block**   The structure of stacked decomposition blocks is shown on the right side of Figure 2. Assuming the input of $l^{th}$ layer is $\mathbf{H}_{\text{data}}^l$ and $\mathbf{H}_{\text{time}}^l$, we can obtain the sub-stable components $\hat{\mathbf{S}}^l \in \mathbb{R}^{n \times k}$ by:

$$\begin{aligned}
\widetilde{\mathbf{H}}^l &= \mathcal{A}_{\text{m}}\left(\mathbf{H}_{\text{data}}^l, \mathbf{H}_{\text{data}}^l, \mathbf{H}_{\text{time}}^l, \mathbf{H}_{\text{time}}^l, \mathbf{H}_{\text{time}}^l\right) \\
\hat{\mathbf{H}}^l &= \text{LayerNorm}\left(\widetilde{\mathbf{H}}^l + \mathcal{A}_{\text{o}}\left(\widetilde{\mathbf{H}}^l, \widetilde{\mathbf{H}}^l, \widetilde{\mathbf{H}}^l\right)\right) \\
\widetilde{\mathbf{S}}^l &= \text{LayerNorm}\left(\hat{\mathbf{H}}^l + \text{FeedForward}\left(\hat{\mathbf{H}}^l\right)\right) \\
\hat{\mathbf{S}}^l &= \text{Out}\left(\widetilde{\mathbf{S}}^l\right)
\end{aligned} \tag{5}$$

where $\widetilde{\mathbf{H}}^l, \hat{\mathbf{H}}^l, \widetilde{\mathbf{S}}^l \in \mathbb{R}^{n \times d_{\text{model}}}$ are intermediate variables. $\text{Out}(\cdot)$ is a single linear layer. Then we can obtain the input of $l+1^{th}$ layer $\mathbf{H}_{\text{data}}^{l+1} = \mathbf{H}_{\text{data}}^l - \widetilde{\mathbf{S}}^l$ and $\mathbf{H}_{\text{time}}^{l+1} = \mathbf{H}_{\text{time}}^l$. Finally, the sub-stable components $\hat{\mathbf{S}}^l$ of all stacked decomposition blocks are summed to obtain the stable component $\hat{\mathbf{S}}$ of the series.

**Offset subtraction**   Although the same time slots in different periods usually have similar fluctuations, it is not strictly one-to-one correspondence. Here, we employ a simple method to handle the horizontal drift, considering the vector $\mathbf{a}$ minus the vector $\mathbf{b}$, with a maximum horizontal offset $d$:

$$\mathcal{M}\left(\mathbf{a}_i, \mathbf{b}\right) = \text{Min}\left(\mathbf{a}_i - \mathbf{b}_{i-d}, \cdots, \mathbf{a}_i - \mathbf{b}_{i+d}\right) \tag{6}$$

where $\mathbf{a}_i$ and $\mathbf{b}_i$ are the $i^{th}$ element in $\mathbf{a}$ and $\mathbf{b}$, respectively. $\text{Min}(\cdot)$ means taking the minimum value. Based on the offset subtraction, we obtain the predicted trend component as $\hat{\mathbf{T}}_{\text{d}} = \mathcal{M}(\mathbf{X}_{\text{d}}, \hat{\mathbf{S}})$.

### 3.4  Diffusion reconstruction

The diffusion reconstruction module mainly consists of a noise diffusion and a reconstruction backbone network. The noise diffusion is used to construct an information bottleneck externally by polluting the input data with noise, and the backbone network is used to reconstruct the polluted data directly.

**Noise diffusion**   Given the original data $x_0$, the polluted data $x_1, x_2, \cdots, x_T$ are obtained by adding $T$ step Gaussian noise. Each moment $t$ in the noise addition process is only relevant to the $t-1$ moment. The hyperparameter $\beta$ controls the ratio of added noise. We add trend retaining to the DDPM so that the model can concentrate on the more crucial stable components. The data $x_t$ after $t$ steps of noise pollution is:

$$x_t = \sqrt{\bar{\alpha}_t}x_0 + \sqrt{1 - \bar{\alpha}_t}\bar{z}_t + \left(1 - \sqrt{\bar{\alpha}_t}\right)b \tag{7}$$

where $\bar{\alpha}_t = \prod_{i=1}^t \alpha_i, \alpha_i = 1 - \beta_i, \bar{z}_t \sim \mathcal{N}(0, 1)$ is the noise, and $b$ is the retained information. The proof is given in Appendix A. So we can obtain the noisy data:

$$\mathbf{X}_{\text{d}}' = \sqrt{\bar{\alpha}_t}\mathbf{X}_{\text{d}} + \sqrt{1 - \bar{\alpha}_t}\bar{\mathbf{Z}} + (1 - \sqrt{\bar{\alpha}_t})\hat{\mathbf{T}}_{\text{d}} \tag{8}$$

where $\mathbf{X}_{\text{d}}' \in \mathbb{R}^{n \times k}, \bar{\mathbf{Z}} \sim \mathcal{N}(0, \mathbf{I})$, and $\bar{\mathbf{Z}} \in \mathbb{R}^{n \times k}$.

Table 1: Statistics of the datasets. A smaller ADF test statistic indicates a more stationary dataset.

| | Training Size | Testing Size | Series Number | Attacks Number | Anomaly Durations | Anomaly Rate | Frequency | ADF Test Statistic |
|---|---|---|---|---|---|---|---|---|
| PSM | 132481 | 87841 | 25 | 73 | 1~8861 | 0.2776 | 1 minute | -9.2314 |
| SMD | 23688 | 23689 | 33 | 30 | 3~3161 | 0.1565 | 1 minute | -4.0947 |
| SWaT | 6840 | 7500 | 25 | 33 | 3~599 | 0.1263 | 1 minute | -2.9442 |

**Backbone network** As shown in the left panel of Figure 2, the backbone network consists of stacked spatial-temporal transformer blocks. The extracted trend $\hat{\mathbf{T}}_d$, the noisy data $\mathbf{X}'_d$, and the timestamp $\mathbf{X}_{\text{time}}$ are encoded by the embedding layer first. To make the network more focused on the stable part without completely ignoring the trend part, we make the hidden vector:

$$\mathbf{H} = \text{embed}(\mathbf{X}'_d) - \text{embed}(\hat{\mathbf{T}}_d) + \text{embed}(\mathbf{X}_{\text{time}}) \tag{9}$$

where $\mathbf{H} \in \mathbb{R}^{n \times d_{\text{model}}}$, $\text{embed}(\cdot)$ is the embedding network. Then, we model the temporal and dimension relationship by feeding $\mathbf{H}$ into the stacked spatial-temporal Transformer blocks. After that, we sum the outputs of the stacked blocks with $\hat{\mathbf{T}}_d$ and send the result into the output layer. Finally, we directly obtain the reconstructed result $\hat{\mathbf{X}}_d$ of $\mathbf{X}_d$ instead of the predicted noise.

### 3.5 Joint optimization

As described in the previous section, the dynamic decomposition and diffusion reconstruction are interconnected. The dynamic decomposition module learns the inherent features of long-period unstable multivariate time series, i.e., the stable component. We use the Mean Square Error (MSE) between $\mathbf{S}$ and $\hat{\mathbf{S}}$ as the loss of this module. The diffusion reconstruction module directly reconstructs the data polluted by noise diffusion. It is worth noting that the output of this module is the reconstruction of $\mathbf{X}_d$. Because the drift $\mathbf{d}$ is added inside $D^3R$, and the external who calculates the loss is only aware of $\mathbf{X}$, we obtain the reconstruction of $\mathbf{X}$ by $\hat{\mathbf{X}} = \hat{\mathbf{X}}_d - \mathbf{d}$, $\hat{\mathbf{X}} \in \mathbb{R}^{n \times k}$. Similar to the dynamic decomposition module, the MSE of $\mathbf{X}$ and $\hat{\mathbf{X}}$ is used directly as the loss of this module.

During the training process, $D^3R$ needs to be trained end-to-end, so the loss function is defined as the sum of the two optimization objectives:

$$Loss = \frac{1}{nk} \sum_{i=1}^{n} \sum_{j=1}^{k} \left( \left( \mathbf{S}_{i,j} - \hat{\mathbf{S}}_{i,j} \right)^2 + \left( \mathbf{X}_{i,j} - \hat{\mathbf{X}}_{i,j} \right)^2 \right) \tag{10}$$

where $\mathbf{S}_{i,j}, \hat{\mathbf{S}}_{i,j}, \mathbf{X}_{i,j}, \hat{\mathbf{X}}_{i,j}$ represent the labeled stable component, the predicted stable component, the original input and the reconstructed output of the $j^{th}$ variable at the $i^{th}$ timestamp, respectively.

### 3.6 Model inference

The online detection (inference) does not need labeled stable component construction and disturbance strategy in the data preprocessing stage. The anomaly score for the current timestamp consists only of the MSE of the original input $\mathbf{X}$ and the reconstructed output. Then, we run the SPOT [23], a streaming algorithm based on extreme value theory, to label each timestamp.

## 4 Experiments

### 4.1 Experimental settings

**Datasets** We evaluate $D^3R$ extensively on three real-world datasets: PSM (Pooled Server Metrics) [1], SMD (Server Machine Dataset) [24], and SWaT (Secure Water Treatment) [15]. For each dataset, we only preserve continuous variables. That is why we do not employ MSL (Mars Science Laboratory rover) and SMAP (Soil Moisture Active Passive satellite) [8], which are discrete except for the first dimension. The normal data is divided into training data (80%) and validation data (20%). Anomaly is only present in the testing data. More descriptions of the datasets are shown in Table 1. Further details about the datasets are available in Appendix B.1.

Table 2: Results in the three real-world datasets. The higher values for all metrics represent the better performance, and the best F1 scores are highlighted in bold.

| Method | PSM | | | SMD | | | SWaT | | | Average |
| | Precision | Recall | F1 | Precision | Recall | F1 | Precision | Recall | F1 | F1 |
| --- | --- | --- | --- | --- | --- | --- | --- | --- | --- | --- |
| Sampling | 0.8439 | 0.5165 | 0.6408 | 0.7453 | 0.3144 | 0.4422 | 0.6062 | 0.8466 | 0.7065 | 0.5965 |
| COPOD | 0.7602 | 0.3175 | 0.4479 | 0.6676 | 0.1366 | 0.2268 | 0.9876 | 0.1180 | 0.2108 | 0.2951 |
| ECOD | 0.7460 | 0.3384 | 0.4656 | 0.7398 | 0.1615 | 0.2651 | 0.9761 | 0.1151 | 0.2059 | 0.3122 |
| OCSVM | 0.8761 | 0.4744 | 0.6155 | 0.0000 | 0.0000 | 0.0000 | 0.6196 | 0.7558 | 0.6810 | 0.4321 |
| PCA | 0.9220 | 0.3771 | 0.5353 | 0.8388 | 0.4019 | 0.5434 | 0.6358 | 0.7218 | 0.6761 | 0.5849 |
| kNN | 0.5317 | 1.0000 | 0.6943 | 0.6988 | 0.3368 | 0.4546 | 0.0000 | 0.0000 | 0.0000 | 0.3830 |
| CBLOF | 0.5990 | 0.9845 | 0.7449 | 0.8667 | 0.3352 | 0.4834 | 0.6308 | 0.7091 | 0.6677 | 0.6320 |
| HBOS | 1.0000 | 0.0654 | 0.1228 | 0.5628 | 0.8007 | 0.6610 | 0.5771 | 0.8049 | 0.6722 | 0.4853 |
| IForest | 1.0000 | 0.0335 | 0.0648 | 1.0000 | 0.0937 | 0.1713 | 0.6127 | 0.6280 | 0.6203 | 0.2855 |
| LODA | 0.9266 | 0.4017 | 0.5605 | 0.5902 | 0.6618 | 0.6240 | 0.6117 | 0.7014 | 0.6535 | 0.6126 |
| VAE | 0.6221 | 0.8772 | 0.7280 | 0.8209 | 0.4349 | 0.5686 | 0.6355 | 0.7218 | 0.6759 | 0.6575 |
| DeepSVDD | 0.7405 | 0.5064 | 0.6015 | 0.6498 | 0.6477 | 0.6488 | 0.5911 | 0.9353 | 0.7244 | 0.6582 |
| LSTM-AE | 0.7511 | 0.7586 | 0.7548 | 0.8496 | 0.4349 | 0.5753 | 0.6018 | 0.7219 | 0.6564 | 0.6622 |
| MTAD-GAT | 0.7990 | 0.6014 | 0.6863 | 0.8590 | 0.6769 | 0.7571 | 0.6590 | 0.7751 | 0.7123 | 0.7186 |
| TFAD | 0.7914 | 0.7163 | 0.7520 | 0.5632 | 0.9783 | 0.7149 | 0.6038 | 0.8196 | 0.6953 | 0.7207 |
| Anomaly Transformer | 0.5201 | 0.8504 | 0.6455 | 1.0000 | 0.0319 | 0.0619 | 0.5541 | 0.5994 | 0.5759 | 0.4278 |
| Adversary | 0.5351 | 0.8971 | 0.6703 | 0.5135 | 0.9663 | 0.6706 | 0.5410 | 0.7531 | 0.6297 | 0.6569 |
| Our | 0.6294 | 0.9619 | **0.7609** | 0.7715 | 0.9926 | **0.8682** | 0.7206 | 0.8529 | **0.7812** | **0.8034** |

**Baselines** We extensively compare our model with 15 baselines, including the probability-based methods: Sampling, COPOD [12], ECOD [13]; the linear transformation-based methods: OCSVM [20], PCA [22]; the proximity-based methods: kNN [18], CBLOF [5], HBOS [4]; the outlier-based methods: IForest [14], LODA [16]; the neural network-based methods: VAE [11], DeepSVDD [19], Lstm-AE [9], MTAD-GAT [33], TFAD [31], Anomaly Transformer [30]. Additionally, we set up an adversary algorithm, which is implemented simply. A timestamp will be marked as abnormal if its sequential id is divisible by $n$, else it will be marked as normal. In our experiment, $n$ is set as 40. This adversary algorithm works as a timed detector and offers no information about the location of the anomaly. All baselines are based on our runs, using the identical hardware. We employ official or open-source implementations published in GitHub and follow the configurations recommended in their papers. Further details concerning the Baselines are available in Appendix B.2.

**Metrics** For performance evaluation, we use precision, recall, and F1 score. The classical metrics are reasonable for tasks with sample granularity but are not applicable for continuous time series, where anomalies are frequently continuous. Most previous work [30, 33, 3, 21] employ the point adjustment method: If a point in a contiguous anomalous segment is detected correctly, all anomalies in the same segment are also considered to have been correctly detected. However, point adjustment is unreasonable, as [10] pointed out, and creates the illusion of progress. Assuming that ground truth and the predicted event are shown in Figure 3, although the real predicted event is just a timed detection, it still achieves an F1 score of 0.91 after point adjustment. The point adjustment may increase TP and decrease FN dramatically [10]. We test all baselines on all datasets with point adjustment

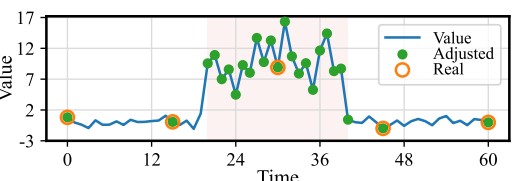

Figure 3: Example of point adjustment strategy. The red area represents the ground truth of the anomaly event. The precision/recall/f1 before adjustment (Real) is 0.20/0.05/0.08, while after adjustment (Adjusted) is 0.84/1.00/0.91.

metrics. The detailed experiment results can be found in Appendix C. Experimental results show that the average F1 score of the adversary algorithm surprisingly outperforms all baselines, even though it can provide no anomaly location information. This phenomenon arises because point adjustment unfairly assigns higher weights to long anomaly events. Furthermore, this algorithm does not consider the adjacency of time series [7]. To address the aforementioned challenges, we employ an F1 score based on affiliation [7]. This score takes into account the average directed distance between predicted anomaly events and ground truth events to calculate precision, as well as the average directed distance between ground truth events and predicted anomaly events to determine recall.

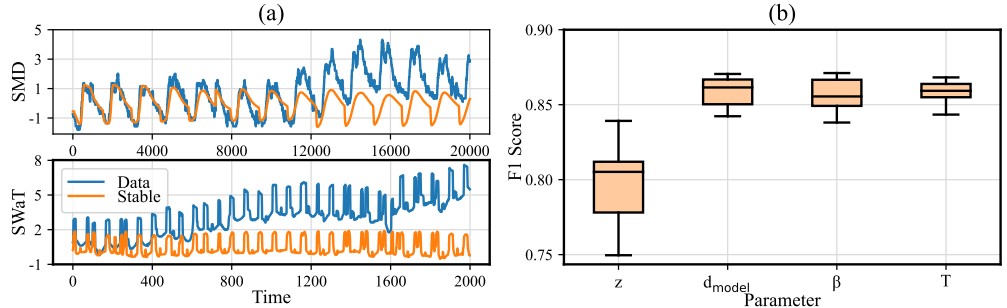

Figure 4: Results of effectiveness analyses. (a) is the visualization in the dynamic decomposition. (b) is the statistic results of sensitivity analyses in the diffusion reconstruction module.

Detailed implementation of D³R can be found in Appendix B.3. All experiments are repeated 5 times, and we report the average for each metric.

### 4.2 Detection Results

On all three real-world datasets, as shown in Table 2, D³R outperforms the adversary algorithm and achieves the best F1 performance, confirming its effectiveness and superiority. Specifically, we achieve 0.61% (0.7548→0.7609), 11.11% (0.7571→0.8682), and 5.68% (0.7244→0.7812) absolute improvement over the previous SOTA methods on PSM, SMD, and SWaT datasets, respectively.

The statistical and machine learning methods frequently perform poorly in generalization because they do not account for time series continuity. They may perform well on a partial dataset while performing poorly or even failing on another (e.g., OCSVM and CBLOF). In opposition, neural network-based methods usually perform more balanced over various datasets. Furthermore, much of the previous works were evaluated by point adjustment, leading to a false boom. The metrics based on affiliation provide a more objective and reasonable evaluation of various methods, although their scores will drop.

Noticeably, D³R significantly outperforms the other methods on the SMD and SWaT datasets, which are typically characterized by high nonstationarity, suggesting the limitation of previous work on unstable real-world data. In our model, the dynamic decomposition and diffusion reconstruction modules complement each other. The former dynamically eliminates the unstable interference from the original data. The latter models the series from more crucial components. These designs tackle the shortcomings of previous work and maintain excellent robustness in complex real-world data.

In addition to the affiliation evaluation metric, we compare D³R with other baselines in all datasets using the AUC score. The detailed experiment results can be found in Appendix D. For a more intuitive comparison, we visualize the anomaly scores of the D³R and partial baselines, as shown in Appendix E. The quantitative and qualitative experimental results all show that D³R outperforms other baselines. Moreover, to validate the practicality of D³R in the actual production environment, we compare run time across different neural network-based algorithms in the SMD and present a summary in Appendix F. Both the training time and inference time of our model are acceptable.

### 4.3 Effectiveness verifications

**Dynamic decomposition module** We propose a dynamic decomposition module to tackle Challenge 1. To more intuitively verify the performance of the dynamic decomposition module, we provide visualization of the decomposition results on highly unstable datasets (SMD and SWaT). Figure 4(a) shows that our model robustly extracts the fundamental stable components, although real-world multivariate data is long-period and complex.

**Diffusion reconstruction module** We propose a diffusion reconstruction module to tackle Challenge 2. To verify the superiority of controlling information bottleneck externally, we replace the entire diffusion reconstruction module with the VAE module and perform sensitivity analyses for the VAE module and diffusion reconstruction module. The hyperparameter that significantly af-

Table 3: Results of ablation studies. F1 scores are reported, with higher values meaning better performance. The best scores are highlighted in bold.

| dataset | D³R | w/o temporal | w/o spatial | w/o time | w/o data | w/o offset | w/o disturbance | w/o diffusion | w/o trend |
|---|---|---|---|---|---|---|---|---|---|
| PSM | **0.7609** | 0.7280 | 0.7571 | 0.7074 | 0.7374 | 0.7420 | 0.7508 | 0.7347 | 0.7592 |
| SMD | **0.8682** | 0.8169 | 0.8173 | 0.7403 | 0.8114 | 0.8446 | 0.6947 | 0.8392 | 0.8006 |
| SWaT | **0.7812** | 0.7031 | 0.7274 | 0.7563 | 0.6720 | 0.7205 | 0.7293 | 0.7293 | 0.7109 |
| Average | **0.8034** | 0.7493 | 0.7673 | 0.7347 | 0.7403 | 0.7690 | 0.7249 | 0.7677 | 0.7569 |

fects the performance of the VAE module is the latent space size $z$. The hyperparameters that may affect the performance of the diffusion reconstruction module are the hidden state dimension $d_{\text{model}}$, the noise ratio $\beta$, and the pollution step $T$. We scale each hyperparameter by a factor of $\{0.2, 0.4, 0.6, 0.8, 1.0, 1.2, 1.4, 1.6, 1.8, 2.0\}$, the statistic results for the SMD are shown in Figure 4(b). As well as allowing adjusting the information bottleneck without retraining, the diffusion reconstruction module is significantly more effective (i.e., the F1 score is higher) and more robust (i.e., the F1 score is more concentrated) than the VAE module. During the hyperparameter changing, the F1 score span of the VAE module is 8.96% (0.7496→0.8392), while the maximum span of the diffusion reconstruction module ($\beta$) is 3.30% (0.8381→0.8711).

## 4.4 Ablation studies

In order to verify the effectiveness and necessity of our designs, we perform ablation studies on both dynamic decomposition and diffusion reconstruction modules of D³R, respectively.

**Dynamic decomposition module** In this module, our main designs are spatial-temporal transformer, data-time mix-attention, offset subtraction, and disturbance strategy. Their ablation results are shown in Table 3. The w/o temporal, w/o spatial, w/o time, w/o data, w/o offset, and w/o disturbance represent the variants of D³R removing temporal transformer, spatial transformer, time attention, data attention, offset subtraction, and disturbance strategy, respectively. The temporal transformer and spatial transformer bring an average absolute improvement of 5.41% (0.7493→0.8034) and 3.61% (0.7673→0.8034), respectively. Compared with the spatial transformer, the temporal transformer is more critical. The spatial transformer performs much worse when the datasets have smaller series number (PSM and SWaT). When the series number is large (SMD), the dimension dependency is more prosperous, and the spatial transformer performs better. Moreover, time attention and data attention in data-time mix-attention bring a great absolute promotion of 6.87% (0.7347→0.8034) and 6.31% (0.7403→0.8034). While data attention is better suited to short-period data (SWaT), time attention is better suited to long-period data (PSM and SMD). Offset subtraction can further improve our model with 3.27% (0.7690→0.8034). Finally, the disturbance strategy brings a significant improvement of 7.85% (0.7249→0.8034), especially for the SMD and SWaT datasets with high nonstationary characteristics.

**Diffusion reconstruction module** This module uses noise diffusion to control the information bottleneck externally. To verify the superiority of noise diffusion, we replace the entire diffusion reconstruction module with the VAE module, and the ablation results are shown in Table 3 (w/o diffusion). As well as allowing adjustment of the information bottleneck without retraining, noise diffusion also provides a 3.57% (0.7677→0.8034) absolute improvement. The following section will discuss more advantages of the diffusion reconstruction module. An additional innovation of this module is trend retaining, and the ablation results are shown in Table 3 (w/o retaining). After removing trend retaining, the performance of D³R decreases significantly, especially in the highly nonstationary SMD dataset (0.8682→0.8006) and SWaT dataset (0.7812→0.7109). Trend retaining enables the model to focus on the critical stable components, avoiding being distracted by irrelevant information.

## 4.5 Hyperparameter analyses

The hyperparameter that significantly affects the performance of the VAE module is the latent space size $z$. The hyperparameters that may affect the performance of D³R are the added drift

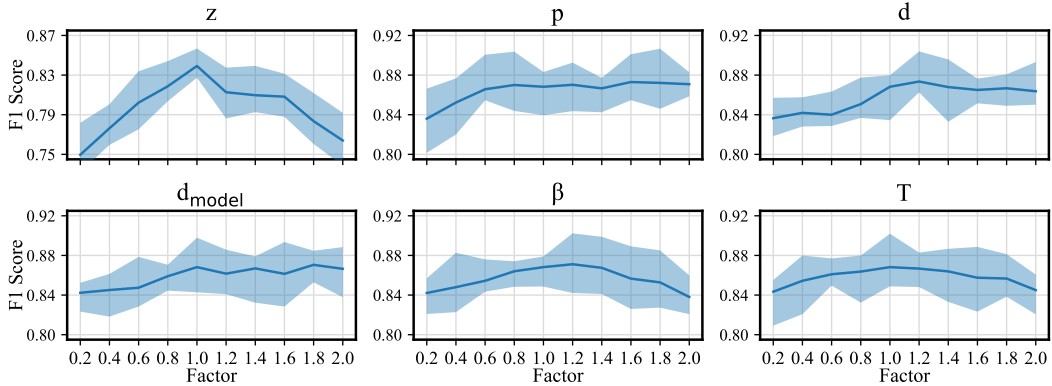

Figure 5: Results of the sensitivity analysis. The higher F1 Score represents the better performance. The dark line represents the mean of 5 experiments, and the light area represents the range.

boundary $p$, the maximum horizontal offset $d$, the hidden state dimension $d_{\text{model}}$, the noise ratio $\beta$, and the pollution step $T$. To analyze their influence on anomaly detection, we perform hyperparameter sensitivity analysis in the SMD. We scale each hyperparameter by a factor of $\{0.2, 0.4, 0.6, 0.8, 1.0, 1.2, 1.4, 1.6, 1.8, 2.0\}$, and the results are shown in Figure 5.

Anomaly detection methods based on reconstruction require constructing the information bottleneck to reconstruct normal data and inhibit anomaly effectively. While traditional autoencoders control the information bottleneck from inside through the latent space size, our approach controls the information bottleneck from outside through noise diffusion. As the experimental results show, with the bottleneck going from strict to loose (latent space size $z$, noise ratio $\beta$, pollution step $T$ going from small to large), the performance of the model goes from up to down. If the information bottleneck is too tight, the normal data cannot be reconstructed, and the precision is low. If the information bottleneck is too loose, the anomaly can be reconstructed well, and the recall is low. On the one hand, our approach to control the information bottleneck externally is more robust (smaller range of variation). On the other hand, as the information bottleneck is not an attribute of the model itself, we can try different bottleneck sizes when inference without retraining the model again.

Since it is no longer an information bottleneck, $D^3R$ is not sensitive to the hidden state dimension $d_{\text{model}}$. A larger hidden space tends to bring better results but also a greater computational cost. As for the added drift boundary $p$ and maximum horizontal offset $d$, the impact on our model becomes small once they exceed a certain threshold. Overall, $D^3R$ is robust to all hyperparameters we test.

## 5   Conclusion

This paper proposes **D**ynamic **D**ecomposition with **D**iffusion **R**econstruction ($D^3R$) for long-period unstable multivariate time series anomaly detection to cover the overlook in previous work. We first decompose the long-period unstable multivariate time series dynamically and then directly reconstruct the data polluted by noise diffusion. Extensive experiments prove that $D^3R$ significantly outperforms existing works. The method we proposed to break the limitation of the local sliding window is also meaningful for other long-period multivariate time series analysis tasks, such as prediction and imputation. Meanwhile, the approach to controlling information bottleneck externally can also be used for anomaly detection of other modal data, such as picture, video, and log data.

## Acknowledgments and Disclosure of Funding

This work was supported in part by the National Natural Science Foundation of China under Grants (62101064, 62171057, 62201072, 62071067), in part by the Ministry of Education and China Mobile Joint Fund (MCM20200202, MCM20180101), in part by the Beijing University of Posts and Telecommunications-China Mobile Research Institute Joint Innovation Center.

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

## A    Noise diffusion proofs

In this section, we explain the proof of Equation 7 in our paper.

Suppose $z \sim \mathcal{N}(0, 1)$ is the Gaussian noise to be added, $\beta$ controls the amount of noise added each step, $\alpha = 1 - \beta$, and $b$ is the trend information to be retained. The $x_t$ after $t$ steps of noise pollution is:

$$\begin{aligned} x_t &= \sqrt{\alpha_t}\left(x_{t-1} - b\right) + \sqrt{1 - \alpha_t}z_t + b \\ &= \sqrt{\alpha_t}x_{t-1} + \sqrt{1 - \alpha_t}z_t + \left(1 - \sqrt{\alpha_t}\right)b \end{aligned} \tag{11}$$

The Gaussian distribution, as we know, has the following properties:

- Suppose $x \sim N(\mu, \sigma^2)$, then $ax + b \sim N(a\mu + b, a^2\sigma^2)$.
- Suppose $x \sim N(\mu_x, \sigma_x^2), y \sim N(\mu_y, \sigma_y^2)$, $x, y$ are independent random variables, then $x + y \sim N(\mu_x + \mu_y, \sigma_x^2 + \sigma_y^2)$.

Based on these properties, the noise diffusion process can be deduced further:

$$\begin{aligned} x_t =& \sqrt{\alpha_t}\left(x_{t-1} - b\right) + \sqrt{1 - \alpha_t}z_t + b \\ =& \sqrt{\alpha_t}\left[\sqrt{\alpha_{t-1}}\left(x_{t-2} - b\right) + \sqrt{1 - \alpha_{t-1}}z_{t-1} + b - b\right] + \sqrt{1 - \alpha_t}z_t + b \\ =& \sqrt{\alpha_t\alpha_{t-1}}x_{t-2} + \sqrt{1 - \alpha_t\alpha_{t-1}}z_t + \left(1 - \sqrt{\alpha_t\alpha_{t-1}}\right)b \end{aligned} \tag{12}$$

Repeating the above process, we can directly infer $x_t$ from $x_0$ as follows:

$$x_t = \sqrt{\bar{\alpha}_t}x_0 + \sqrt{1 - \bar{\alpha}_t}\bar{z}_t + \left(1 - \sqrt{\bar{\alpha}_t}\right)b \tag{13}$$

where $\bar{\alpha}_t = \prod_{i=1}^{t} \alpha_i$ and $\bar{z}_t \sim \mathcal{N}(0, 1)$.

## B    Detailed experimental settings

### B.1    Datasets

The datasets can be downloaded from the following:

- PSM: https://github.com/eBay/RANSynCoders/tree/main/data.
- SMD: https://github.com/NetManAIOps/OmniAnomaly/tree/master/ServerMachineDataset.
- SWaT: https://itrust.sutd.edu.sg/testbeds/secure-water-treatment-swat/.

### B.2    Baselines

All baselines are based on our runs, using the identical hardware. We employ official or open-source implementations published in GitHub and follow the configurations recommended in their papers. The baselines can be downloaded from the following:

- Sampling: https://github.com/yzhao062/pyod.
- COPOD: https://github.com/yzhao062/pyod.
- ECOD: https://github.com/yzhao062/pyod.
- OCSVM: https://github.com/yzhao062/pyod.
- PCA: https://github.com/yzhao062/pyod.
- kNN: https://github.com/yzhao062/pyod.
- CBLOF: https://github.com/yzhao062/pyod.
- HBOS: https://github.com/yzhao062/pyod.
- IForest: https://github.com/yzhao062/pyod.
- LODA: https://github.com/yzhao062/pyod.
- VAE: https://github.com/yzhao062/pyod.

Table 4: Quantitative results after point adjustment in the three real-world datasets. The higher values for all metrics represent the better performance, and the best F1 scores are highlighted in bold.

| Method | PSM | | | SMD | | | SWaT | | | Average |
|---|---|---|---|---|---|---|---|---|---|---|
| | Precision | Recall | F1 | Precision | Recall | F1 | Precision | Recall | F1 | F1 |
| Sampling | 0.9986 | 0.8571 | 0.9224 | 0.9958 | 0.9690 | 0.9822 | 0.1583 | 0.9271 | 0.2705 | 0.7250 |
| COPOD | 0.9795 | 0.8572 | 0.9143 | 0.9975 | 0.8538 | 0.9201 | 0.9953 | 0.6695 | 0.8005 | 0.8783 |
| ECOD | 0.9565 | 0.8632 | 0.9075 | 0.9645 | 0.8573 | 0.9078 | 0.9954 | 0.6832 | 0.8103 | 0.8752 |
| OCSVM | 0.9990 | 0.8115 | 0.8955 | 0.0000 | 0.0000 | 0.0000 | 0.1596 | 0.9113 | 0.2716 | 0.3890 |
| PCA | 0.9996 | 0.7777 | 0.8748 | 0.9964 | 0.9720 | 0.9840 | 0.1645 | 0.9029 | 0.2783 | 0.7124 |
| kNN | 0.2776 | 1.0000 | 0.4345 | 0.9950 | 0.9676 | 0.9811 | 0.0000 | 0.0000 | 0.0000 | 0.4719 |
| CBLOF | 0.4301 | 0.9996 | 0.6014 | 0.9970 | 0.9698 | 0.9832 | 0.1635 | 0.9029 | 0.2769 | 0.6205 |
| HBOS | 1.0000 | 0.5957 | 0.7466 | 0.3745 | 0.9922 | 0.5437 | 0.3932 | 0.8965 | 0.5467 | 0.6123 |
| IForest | 1.0000 | 0.2204 | 0.3612 | 1.0000 | 0.8560 | 0.9224 | 0.4204 | 0.8141 | 0.5545 | 0.6127 |
| LODA | 0.9975 | 0.8465 | 0.9158 | 0.8452 | 0.9803 | 0.9077 | 0.2502 | 0.8912 | 0.3907 | 0.7381 |
| VAE | 0.6020 | 0.9486 | 0.7365 | 0.9953 | 0.9720 | 0.9835 | 0.1646 | 0.9029 | 0.2784 | 0.6661 |
| DeepSVDD | 0.6919 | 0.8779 | 0.7739 | 0.9575 | 0.9714 | 0.9644 | 0.1465 | 0.9842 | 0.2551 | 0.6645 |
| LSTM-AE | 0.7280 | 0.9288 | 0.8162 | 0.9961 | 0.9720 | 0.9839 | 0.1614 | 0.9029 | 0.2739 | 0.6913 |
| MTAD-GAT | 0.9589 | 0.8987 | 0.9279 | 0.9940 | 0.9857 | **0.9898** | 0.1745 | 0.9197 | 0.2934 | 0.7370 |
| TFAD | 0.8819 | 0.9851 | 0.9306 | 0.9991 | 0.9492 | 0.9735 | 0.2559 | 0.9387 | 0.4022 | 0.7688 |
| Anomaly Transformer | 0.9022 | 0.9879 | 0.9431 | 1.0000 | 0.8525 | 0.9204 | 0.9076 | 0.7159 | 0.8005 | 0.8880 |
| Adversary | 0.9384 | 0.9921 | **0.9645** | 0.8774 | 0.9649 | 0.9191 | 0.8309 | 0.8300 | **0.8304** | **0.9047** |

- DeepSVDD: https://github.com/yzhao062/pyod.

- LSTM-AE: https://github.com/matanle51/LSTM_AutoEncoder.

- MTAD-GAT: https://github.com/ML4ITS/mtad-gat-pytorch.

- TFAD: https://github.com/DAMO-DI-ML/CIKM22-TFAD.

- Anomaly Transformer: https://github.com/thuml/Anomaly-Transformer.

## B.3 Implementation

In our experiment, the sliding window has a fixed size 64 for all datasets. We set the dimension of hidden states as 512, the dimension of attention as 64, the number of attention heads as 8, the number of network layers as 2, and the dropout as 0.6. We use grid search to obtain the best SPOT parameters for each dataset and record the results with the highest F1 scores. The $\beta$ of the diffusion model changes from 0.0001 to 0.02 in 1000 steps like DDPM without adjustment. We add noise with 500 steps (selected from $\{100, 300, 500, 700, 900\}$) as an external information bottleneck when the noise diffusion. The boundary of added drift in the disturbance strategy is 10 (selected from $\{1, 2.5, 5, 10, 20\}$). The max offset of offset subtraction is 30 (selected from $\{5, 10, 20, 30, 40\}$). We use the Adam optimizer with an initial learning rate of $10^{-4}$. The training process is early stopped within 8 epochs with a batch size of 8. The implementation of $D^3R$ is carried out using Python 3.9.13 and PyTorch 1.11.0. All experiments are performed on a Ubuntu Server with a 12th Gen Intel(R) Core(TM) i9-12900K @ 3.60GHz processor and an NVIDIA GeForce RTX 3090 Graphics Card.

## C Detection results with point adjusted

We test all baselines on all datasets with point-adjusted evaluation metrics, and the detailed quantitative results are shown in Table 4. Since the data processing and hardware are not the same as those in the original paper, the results of the baselines in our experiments are slightly different from those in the original paper. Nevertheless, the errors are still within a reasonable range.

The point adjustment method is unreasonable. Despite the no helpful information provided by the adversary algorithm, its average performance still exceeds other SOTA methods by 2% of the relative F1 score. Specifically, the adversary algorithm achieves an absolute improvement of 2.14% (0.9431→0.9645) and 2.01% (0.8103→0.8304) on the PSM and SWaT datasets, respectively. Only on the SMD dataset the adversary algorithm is slightly weaker than the best baseline (0.9898→0.9191). If we further adjust the $n$ in the adversary algorithm, it can also outperform all the baselines in the SMD dataset.

Table 5: AUC score in the three real-world datasets. The higher AUC score represents the better performance, and the best AUC scores are highlighted in bold.

| Method | PSM | SMD | SWaT | Average |
|---|---|---|---|---|
| Sampling | 0.8791 | 0.9611 | 0.5348 | 0.7917 |
| COPOD | 0.8526 | 0.9230 | 0.7396 | 0.8384 |
| ECOD | 0.8394 | 0.9220 | 0.7532 | 0.8382 |
| OCSVM | 0.8708 | 0.6789 | 0.5379 | 0.6959 |
| PCA | 0.8617 | 0.9671 | 0.5324 | 0.7871 |
| kNN | 0.8640 | 0.8935 | 0.5806 | 0.7793 |
| CBLOF | 0.8681 | 0.9694 | 0.5378 | 0.7918 |
| HBOS | 0.8150 | 0.7393 | 0.8085 | 0.7876 |
| IForest | 0.8892 | 0.9218 | 0.7238 | 0.8450 |
| LODA | 0.8619 | 0.9180 | 0.6780 | 0.8193 |
| VAE | 0.8583 | 0.9674 | 0.5325 | 0.7861 |
| DeepSVDD | 0.8100 | 0.9187 | 0.5063 | 0.7450 |
| LSTM-AE | 0.8894 | 0.9698 | 0.6255 | 0.8283 |
| MTAD-GAT | 0.9093 | 0.9443 | 0.6386 | 0.8307 |
| TFAD | 0.8185 | 0.9386 | 0.6966 | 0.8179 |
| Anomaly Transformer | 0.7074 | 0.7150 | 0.6638 | 0.6954 |
| Adversary | 0.7199 | 0.5374 | 0.6487 | 0.6353 |
| Our | **0.9223** | **0.9759** | **0.8554** | **0.9179** |

# D  Additional detection results

In order to avoid the influence of SPOT parameters and thresholds on the results, we compare the methods from the perspective of the original anomaly scores. If an anomaly lasts too long, more weight will be assigned, resulting in an inaccurate metric. Thus, we aggregate each anomaly event. An anomaly event after aggregation is equivalent to only one timestamp, labeled as an anomaly, and scored as the maximum of the original anomaly range.

We use Area Under Curve (AUC) to evaluate the anomaly scores of each method. The value of AUC ranges between 0.5 and 1, while the closer to 1.0, the better the method. The results of the experiments are shown in Table 5. $D^3R$ significantly outperforms existing methods, with a 9% average relative improvement over other SOTA methods. Specifically, we achieve 1.30% (0.9093→0.9223), 0.61% (0.9698→0.9759), and 4.69% (0.8085→0.8554) absolute improvement over the best SOTA performance on PSM, SMD and SWaT datasets, respectively. This experiment proves the superiority of $D^3R$ once again.

# E  Anomaly score visualizations

We visualize the anomaly scores of the partial baseline and $D^3R$ for a more intuitive comparison, as shown in Figure 6. $D^3R$ consistently provided the most distinguishable anomaly scores in all three real-world datasets. In the PSM dataset, the vertical drift is insignificant. In this case, some of the baselines (MTAD-GAT) can achieve correct detection results, but there are still missed detections (Anomaly Transformer) and false detections (VAE). In the SMD and SWaT datasets, the vertical drift of the data is significant. The anomaly scores of baselines in the normal region are already far from the training data. The unstable normal data is likely to be mistakenly detected as an anomaly. Once the anomaly occurs, even if the anomaly scores of the baselines improve, they are not significant enough due to the large base.

# F  Run Times

We comprehensively compare training time, inference time, and model size for neural network-based models on the SMD dataset to validate the practicality of $D^3R$ in the production environment. The outcomes are presented in Table 6.

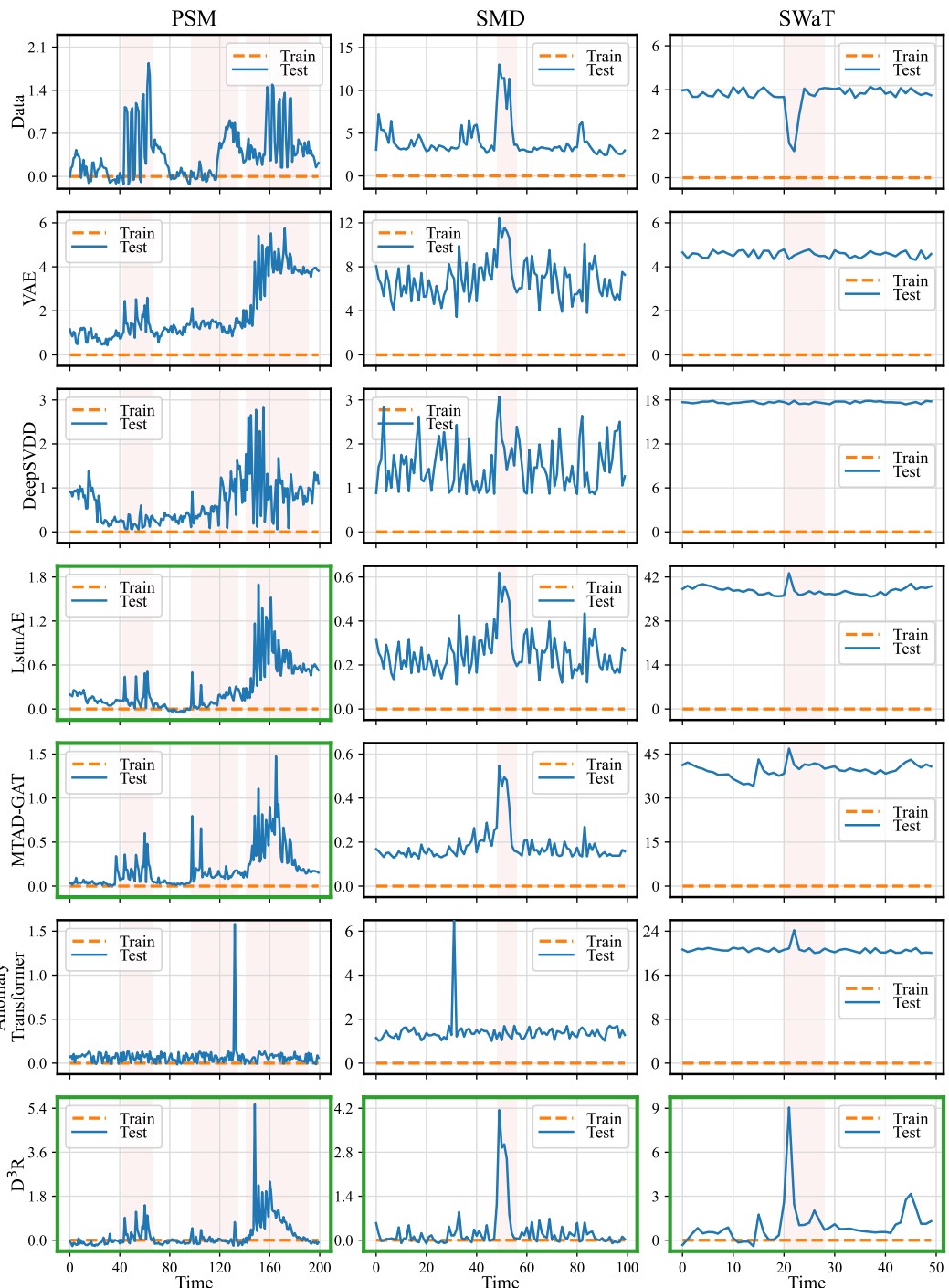

Figure 6: Visualization of the anomaly scores in the three real-world datasets. The red area represents the ground truth of the anomaly event. In the first row, we visualize the raw data. In the remaining six rows, we visualize the anomaly scores provided by different methods. The moment with a higher anomaly score is more likely to be identified as an anomaly. Green boxes bound the successful cases. We normalize the data according to the statistical characteristics of the training set. Test denotes the actual value of the testing set after the normalization, while Train denotes the average value of the training set after the normalization, which is always 0.

Table 6: Run times in the SMD. The lower values represent the better performance. The best results are highlighted in bold, and the worst results are underlined.

| Method | Training Time (s) | Inference Time (s) | Model Size (MB) |
|---|---|---|---|
| VAE | 157.91 | 30.90 | 0.02 |
| DeepSVDD | **60.70** | **12.61** | **0.01** |
| LSTM-AE | 283.61 | 72.82 | 0.04 |
| MTAD-GAT | 188.52 | 60.23 | 1.20 |
| TFAD | 315.39 | 38.54 | 1.04 |
| Anomaly Transformer | 422.43 | 94.36 | 28.15 |
| Our | 399.32 | 104.12 | 109.35 |

Both the training set and the test set of SMD have approximately 16 days of data. Statistically, the training time for these models remains below 10 minutes, which is acceptable for real-world deployment and maintenance. The utilization of attention and its variants in the backbone network of our model leads to its larger size compared to previous algorithms. Thanks to the highly parallelized nature of the attention mechanism, both the training time and inference time of our model remain competitive. The inference process for 16 days of data necessitates a mere 104.12 seconds, satisfying the online, real-time detection criteria. Compared to the substantial enhancement (11%) in detection accuracy that we have achieved, the model size of 109.35MB remains affordable within the context of the expanding hardware resources of the present era. Additionally, there has been substantial recent work (such as Flowformer) on transformer linearization, which may aid in reducing the burden of our model. We plan to explore this aspect in our future research.

## G  Broader Impacts

Recently, Deep Learning has been increasingly used in anomaly detection. It improves safety and prevents potential risks and financial losses by detecting anomalies in healthcare, industrial manufacturing, and autonomous driving. However, since real-time data is constantly changing, data characteristics can change drastically as time accumulates. Due to the obsolescence of the training data, our models cannot adapt to this situation and may provide incorrect judgments. In the future, we consider designing algorithms that adopt online updates to address this shortcoming.

## H  Limitations

The computational cost of our model is large due to the Transformer architecture used for the backbone network. In future work, we plan to replace it with a more lightweight and efficient base structure, such as dilated convolutions and graph neural networks. Furthermore, due to the shortcomings of the point-adjustment evaluation metric, in order to prove the superiority of $D^3R$, we have to compare various other metrics and provide rich visualizations. We plan to design a more intuitive and effective anomaly detection evaluation metric in future work.

