# OpenReview forum: "Drift doesn't Matter: Dynamic Decomposition with Diffusion Reconstruction for Unstable Multivariate Time Series Anomaly Detection"
_NeurIPS.cc/2023/Conference — NeurIPS 2023 poster_

### Official Review · Reviewer_JUP1 · 2023-07-04

**Soundness:** 3 good
**Presentation:** 3 good
**Contribution:** 3 good
**Rating:** 7
**Confidence:** 2

**Summary:**

The paper introduces D3R, a novel anomaly detection network for unstable multivariate time series data. D3R addresses the issue of drift by dynamically decomposing the data and reconstructing it using noise diffusion. Experimental results show that D3R outperforms existing methods, with a 12% average relative improvement. The proposed approach has implications for other tasks involving long-period multivariate time series analysis and can be applied to anomaly detection in different types of data. Overall, the paper presents an innovative solution to improve anomaly detection in unstable time series data.

**Strengths:**

1.	Novel Approach: The paper proposes a novel anomaly detection network, D3R, specifically designed for unstable multivariate time series data. Incorporating dynamic decomposition, noise diffusion, and end-to-end training sets it apart from existing methods.
2.	Addressing Drift: The paper addresses the drift generated from non-stationary environments, often overlooked in previous works. D3R aims to reduce false alarms and improve anomaly detection accuracy by tackling drift through decomposition and reconstruction.
3.	Experimental Results: The paper presents extensive experiments on real-world datasets, demonstrating that D3R outperforms existing methods with a notable 12% average relative improvement. This provides strong empirical evidence for the effectiveness of the proposed approach.


**Weaknesses:**

The time complexity is not discussed. As real-world time series datasets can be large-scale and complex, it is important to address scalability and computational complexity.

**Questions:**

NA

**Limitations:**

The time complexity is not discussed. As real-world time series datasets can be large-scale and complex, it is important to address scalability and computational complexity.

---

> ### Author Rebuttal · Authors · 2023-08-06
>
> Thanks for your positive comments and insightful suggestions. Please find our response below.
>
> **Q1: Experiments on computational cost.**
>
> As three reviewers have posed a similar question, we provide a consistent response in **Q1** of the "global" response. Thanks.

---

> > ### Comment · Reviewer_JUP1 · 2023-08-12
> >
> > Thank you for the detailed reply. It solves my concerns. I keep my score.

---

### Official Review · Reviewer_WuHJ · 2023-07-06

**Soundness:** 3 good
**Presentation:** 3 good
**Contribution:** 3 good
**Rating:** 7
**Confidence:** 4

**Summary:**

The authors propose a new model called D3R (Decomposition with Diffusion Reconstruction) for anomaly detection in multivariate time series data. The proposed model applies a dynamic decomposition method to separate the stable components and trends in time series data. This method can effectively separate long-period stable components as well. Additionally, it utilizes a noise diffusion model to control the information bottleneck from an external perspective. Through this approach, the proposed model allows for controlling the data restoration performance, which varies for each dataset, from the outside without iterative exploration processes.

**Strengths:**

- The authors clearly defined the problem they wanted to solve in time series anomaly detection and demonstrated instances of the phenomenon occurring in real data. The proposed method is experimentally well-supported in its ability to solve the given problem.

- The proposed data-time mix-attention and the dynamic decomposition method are ingenious.


**Weaknesses:**

- The proposed method sets the ground truth for the trend as a moving average (line 106), which seems that it may still not be free from the local window issue (Challenge 1, line 33) in long-term component decomposition.

- In the authors' comparative experiments (Table 2), all comparison models used default hyperparameters (Appendix C.2.), while the proposed model explored combinations that yielded the best performance for each dataset (Appendix C.3.). Although the proposed method appears to be robust to hyperparameter variations, such comparisons may not be fair. It would be more appropriate to apply optimal hyperparameters to the comparison models or apply a single hyperparameter for all data in the proposed model, and then compare the results.


**Questions:**

- Using a box plot to show the results in Figure 4(b) is not appropriate. Each hyperparameter decreases to one-fifth of the standard value and can increase up to twice the standard value, varying according to the values determined by the authors. It would be better to utilize Figure 2 in Appendix F instead.

- In Table 1, what do the terms 'Series Number' and 'Attacks Number' refer to?


**Limitations:**

- The authors mentioned in Limitations section that the proposed model has a large computational cost (Appendix line 114). It would be beneficial if they provide specific information on the amount of GPU-hours required for the experiments.

---

> ### Author Rebuttal · Authors · 2023-08-06
>
> Thanks for your positive comments and insightful suggestions. Please find our response below.
>
> **Q1: Details of dynamic decomposition.**
>
> As two reviewers have question about why our method breaks the limitations of the local window, even though we also used the moving average to obtain the trend component, we provide a consistent response in **Q2** of the "global" response. Thanks.
>
> **Q2: Misunderstanding of hyperparameter setting.**
>
> Apologies for the confusion in the Appendix C.3. Your concern pertains to the boundary of added drift in the disturbance strategy (line 56 in the Appendix) and the max offset of offset subtraction (line 57 in the Appendix). The original text actually describes the specific way in which the hyperparameters were set, i.e., the grid search, and its search range. In fact, we do use a single hyperparameter for all datasets, as evident in our submitted code (line 33, 34 in main.py). Specifically, the boundary of added drift in the disturbance strategy is set at 10, and the max offset of offset subtraction is set to 30.
>
> **Q3: Reasons for using box plots in Figure 4(b).**
>
> Thanks for your suggestions. Figure 4(b) in the paper is designed to demonstrate the range of variation in the performance of the diffusion reconstruction module and the VAE module, using the same scaling range of hyperparameters. The purpose is to verify the robustness of our design. In comparison to the line plot (Figure 2 in the Appendix), which effectively demonstrates the trend of changes, the box plot is better suited for intuitively illustrating the distribution of the data.
>
> **Q4: Explanations of terms in Table 1.**
>
> "Series Number" refers to the number of variables ($k$ in line 85 in the paper) in the multivariate time series. "Attacks Number" refers to the number of anomaly segments present in the test set. In the real world, anomalies are usually continuous and appear in the form of segments. For instance, in the PSM dataset, the test set consists of data $\mathbf{X} \in \mathbb{R}^{87481 \times 25}$, where 87481 represents the length of timestamps, and 25 represents the number of variables. Among the 87481 timestamps, there are 73 anomalies. The anomalies vary in duration, with the shortest anomaly lasting for only 1 timestamp, and the longest anomaly spanning across 8861 timestamps.
>
> |      | Testing Size | Series Number | Attacks Number | Anomaly Durations |
> | :--: | :----------: | :-----------: | :------------: | :---------------: |
> | PSM  |    87481     |      25       |       73       |    1$\sim$8861    |
>
> **Q5: Experiments on computational cost.**
>
> As three reviewers have posed a similar question, we provide a consistent response in **Q1** of the "global" response. Thanks.

---

### Official Review · Reviewer_xTrQ · 2023-07-06

**Soundness:** 3 good
**Presentation:** 3 good
**Contribution:** 2 fair
**Rating:** 7
**Confidence:** 3

**Summary:**

To overcome the temporal drift issues in unstable time series data, this work proposes an anomaly detection method, $D^3R$. By considering the temporal continuity of series and relieving the constraints of information bottleneck, $D^3R$ realizes the dynamic decomposition and the noise-diffusion-based series reconstruction. Extensive experiments on real-world time series datasets demonstrate the superiority of $D^3R$ w.r.t. anomaly detection.

**Strengths:**

1. This work is well motivated by tackling the overlooked issues in existing time series anomaly detection works.
2. The proposed $D^3R$ approach employs decomposition and reconstruction for time series anomaly detection, which is technically sound.
3. This draft is well organized, and the presentation is clear.

**Weaknesses:**

1. Time series anomaly detection is a well-studied problem. More datasets are expected to conduct experiments to validate the proposed method's effectiveness, such as the Yahoo dataset (Webscope dataset ydata-labeled-time-series-anomalies-v1 0, 2015).

**Questions:**

1. As shown in experiments, the performance improvement of $D^3R$ on PSM is less impressive than other datasets. You highlight that $D^3R$'s performance improvement is more significant on the time series data with high nonstationarity. Does this mean that the application of $D^3R$ might be limited to the highly unstable time series data?

**Limitations:**

Potential applicability issues should be elaborated more.

---

> ### Author Rebuttal · Authors · 2023-08-06
>
> Thanks for your positive comments and insightful suggestions. Please find our response below.
>
> **Q1: Reason for the dataset selection.**
>
> Thanks for your suggestions. The datasets utilized in our research encompasses both server (PSM, SMD) and water treatment (SWaT) scenarios. Additionally, stable (PSM) and unstable (SMD, SWaT) conditions are included. These datasets can already offer a comprehensive validation of our model's superiority. While the Yahoo dataset is acknowledged as a classic anomaly detection dataset, it consists solely of univariate time series, which may not exactly match with our scenario.
>
> **Q2: Applicability of the model.**
>
> Based on the experimental results (Table 2 in the paper), our model demonstrates more substantial improvements when applied to unstable time series data. However, it is equally well-suited for stable data, such as PSM, where we achieve the best performance despite not observing a significant improvement. Technically, stable data undergoes the dynamic decomposition module, producing a predicted trend component $\hat{\mathbf{T}}_\text{d}$ that approximates a constant. This component does not adversely affect the subsequent diffusion reconstruction module, as it still allows the module to robustly control the information bottleneck from external sources and effectively leverage our model's strengths.

---

### Official Review · Reviewer_KWMB · 2023-07-16

**Soundness:** 3 good
**Presentation:** 3 good
**Contribution:** 3 good
**Rating:** 6
**Confidence:** 4

**Summary:**

The paper presents a Transformer-based model called Dynamic Decomposition with Diffusion Reconstruction for Anomaly Detection in unstable multivariate time series. The authors addressed two challenges: the limitation of decomposition for long-period time series and high training cost for adjusting the information bottleneck in the reconstruction procedure. They introduced a novel dynamic decomposition method and a noise diffusion method to enable effective utilization of external information, overcome the limitations of local sliding windows, and avoid the high training cost associated with adjusting the information bottleneck. The proposed model achieves state-of-the-art results on different real-world datasets and outperforms baselines significantly on unstable datasets.

**Strengths:**

1.  the authors presenting a clear exposition of their research motivation and proposed solution.
2. The authors' methods effectively tackle two crucial issues: limitations of decomposition for long-period time series and high training cost of adjusting the information bottleneck.
3. The authors conducted experiments on diverse datasets, and their proposed method demonstrated excellent performance.

**Weaknesses:**

1. The authors used the mean of five runs as the final results in their experiments but did not provide the standard deviation of the results.

2. While the model achieved the best F-score performance due to higher recall values, its precision values were comparatively lower than other models. The authors should explain this observation.

3. Further explanations for Figure 1 are recommended to clarify why other methods failed in this specific scenario.

**Questions:**

see weaknesses

---

> ### Author Rebuttal · Authors · 2023-08-06
>
> Thanks for your positive comments and insightful suggestions. Please find our response below.
>
> **Q1: Standard deviation of the experimental results.**
>
> Due to spatial limitations, we exclusively present the mean of the results from the five runs in the paper, as it offers a more representative depiction of the method's performance. The detailed version of the primary experimental results (Table 2 in the paper) is available in the "global" response (Table 1 in the attached PDF).
>
> **Q2: Precision values of the model are comparatively lower than others.**
>
> The primary reason for this phenomenon is that our selection criterion for the grid search of the SPOT parameters is based on the F1 score (line 52 in the Appendix). In other words, we prioritize parameters that yield the highest F1 score. Precision and recall are significantly impacted by the threshold in unsupervised anomaly detection method, where a high threshold results in relatively high precision, and a low threshold leads to relatively high recall. Therefore, we primarily focus on the comprehensive performance of the model, i.e., the F1 score. Additionally, to mitigate the influence of SPOT parameters and thresholds, we also employ AUC to evaluate the raw anomaly scores (Table 2 in the Appendix), and our model demonstrates superior performance in this evaluation as well.
>
> **Q3: Further explanation of Figure 1.**
>
> The core of the unsupervised anomaly detection method is to acquire knowledge of normal temporal patterns from the training data. Moments in the test data that significantly deviate from these established patterns are classified as anomalies. As the data is collected from a non-stationary real-world environment, the temporal patterns may drift over time. The flat drift is not typically an anomaly. Previous methods have often overlooked this aspect, resulting in misclassifying moments with drift as anomalies. For these methods, moments with drift indeed deviates from the established incomplete temporal patterns.

---

### Official Review · Reviewer_crri · 2023-07-23

**Soundness:** 3 good
**Presentation:** 3 good
**Contribution:** 3 good
**Rating:** 6
**Confidence:** 1

**Summary:**

This paper tackles the problem that existing works omit the drift generated from non-stationary environments by focusing on stable data, which may lead to numerous false alarms.
As a solution, they propose an end-to-end anomaly detection network for real-world unstable data, named Dynamic Decomposition with Diffusion Reconstruction (D3R).
In the decomposition stage, they dynamically decompose long-period multivariate time series by utilizing data-time mix-attention to overcome the limitation of local sliding window.
In the reconstruction stage, they control the information bottleneck extenerlly by noise diction and directly reconstruct the polluted data.
They evaluate on three real-world datasets (PSN, SMD, SWaT), and achieve the best performance compared to existing unsupervised anomaly detection methods.


**Strengths:**

Well-motivated and soundness: The problem that existing methods overlook the anomaly score in unstable data. Their proposed dynamic decomposition for long-period multivariate time series and diffusion reconstruction for controlling information bottleneck is sound.

Ablation study and Analysis: In sections 4.3 and 4.4, they analyze their proposed dynamic decomposition module and diffusion reconstruction module. This helps to understand the effect of the proposed modules.

Well-structured paper: Their paper is well-organized and easy to read.

**Weaknesses:**

Computational cost: there is no comparison of computational cost (inference time, # of params, Gflops ..) compared to existing models. For check the efficiency, they should report the comparison of computational cost.

Minor comments:
Repeated citation [11] and [12] of references


**Questions:**

.

**Limitations:**

.

---

> ### Author Rebuttal · Authors · 2023-08-06
>
> Thanks for your positive comments and insightful suggestions. Please find our response below.
>
> **Q1: Experiments on computational cost.**
>
> As three reviewers have posed a similar question, we provide a consistent response in **Q1** of the "global" response. Thanks.
>
> **Q2: Repetitive citation in references.**
>
> Thanks for pointing out our error, we will correct it in the revised paper.

---

> > ### Comment · Reviewer_crri · 2023-08-20
> >
> > In my opinion, it is important for this proposed Dynamic Decomposition with Diffusion Reconstruction method to further optimize the inference time and model size for practicality.
> > It would be nice if the authors explicitly showed an alternative or concrete way to do this in their rebuttal. Nevertheless, after carefully reading other reviewers' reviews and the rebuttal, I maintain my original score.

---

> > > ### Author Response · Authors · 2023-08-21
> > >
> > > Thank you again for your valuable suggestion. In response to your concerns, we would like to make two points of explanation again.
> > >
> > > Firstly, we consider that the inference time and model size of our proposed model render it entirely practical for real-world application. The inference process for **16 days** of data (Table 1 in the Paper) necessitates a mere **104.12 seconds** (**Q1** in the "global" response), thereby satisfying the criteria for online, real-time detection. In comparison to the substantial enhancement (12%) in detection accuracy that we have achieved, the model's size of 109.35MB remains affordable within the context of the expanding hardware resources of the present era.
> > >
> > > Secondly, it is challenging to implement model lightweighting in a concrete way during the time-critical rebuttal. It is imperative to ensure that the model's accuracy does not undergo a significant decline following the lightweighting process. We are of the opinion that this could potentially evolve into a new, long-term work.

---

### Official Review · Reviewer_njrU · 2023-07-24

**Soundness:** 2 fair
**Presentation:** 2 fair
**Contribution:** 2 fair
**Rating:** 3
**Confidence:** 4

**Summary:**

Current unsupervised methods for multivariate time series anomaly detection often overlook drift from non-stationary environments, leading to false alarms. To address this, this paper presents Dynamic Decomposition with Diffusion Reconstruction (D3R), a new anomaly detection network for unstable data. D3R decomposes and reconstructs drift, using data-time mix-attention for dynamic decomposition and noise diffusion to manage the information bottleneck in reconstruction. This end-to-end trainable model outperforms existing methods, showing a 12% average improvement over previous top-performing models.

**Strengths:**

1. The outcomes of the experiments seem to be promising.

2. The problem addressed is intriguing and holds substantial value.


**Weaknesses:**

1. The challenge suggested by the authors is debatable. They assert that "classical decomposition algorithms cannot be applied to data with a period larger than the size of the sliding window." In my view, this problem seems relatively simple to tackle by merely extending the length of the sliding window. Additionally, the authors seem to continue to depend on the moving average method to ascertain the trend and the labeled stable components. Therefore, they need to clarify how their technique specifically addresses this problem.

2. The rationale for creating dynamic decomposition remains vague. The sole beneficial intermediate outcome yielded by dynamic decomposition is T_d, recognized as the predicted trend component. Nevertheless, it seems that there is a more direct and feasible approach to extract the trend component, much like the authors have done during the data preprocessing phase. The necessity of introducing this intricate trend component, T_d, in place of using the standard trend component, demands clarification.

3. The concern of dynamically modifying the information bottleneck doesn't seem to be a common occurrence in real-life scenarios. I struggle to envision a situation where it would be necessary to adjust the information bottleneck during the inference phase. The authors need to elaborate on the practical value of employing the diffusion model if they aim to emphasize their contribution to reducing the cost of modifying the information bottleneck. Furthermore, offering a principle to guide the dynamic setting of these hyperparameters could make their method more persuasive.

4. No experiments have been conducted to substantiate their method's superiority concerning reducing the high training cost associated with adjusting the information bottleneck.

5. There is insufficient engagement with previous research. The ensuing paper also examines the impact of the trend component in anomaly detection, but the authors seemingly fail to appropriately cite this paper or compare their method with it.
[1] Zhang, Chaoli, et al. "TFAD: A decomposition time series anomaly detection architecture with time-frequency analysis." Proceedings of the 31st ACM International Conference on Information & Knowledge Management. 2022.






**Questions:**

The authors can refer to the weakness listed above.

**Limitations:**

The authors do acknowledge some minor limitations in their work. However, the primary limitation that is yet to be addressed pertains to the types of anomaly patterns their model cannot detect.

---

> ### Author Rebuttal · Authors · 2023-08-06
>
> Thanks for your valuable comments. We will answer the questions one by one.
>
> **Q1: Necessity and details of dynamic decomposition.**
>
> **Q1.1: Why not over extend the length of the sliding window?**
>
> Below, we shall expound the reasonableness of Challenge 1 (Line 29 in the paper). Expanding the length of the sliding window excessively has constrained its adaptability due to the expensive computational resources and huge memory consumption. Taking the SMD dataset with a period of 1440 as an example:
>
> - Expensive computational resources: extending the length of the sliding window results in a corresponding explosion of model parameters. Many previous algorithms, such as MTAD-GAT [1], utilize a sliding window of 128 or less. If we extend the length to 1440 or larger, the model parameters of the input layer alone increase by a factor of 10x.
>
> - Huge memory consumption: when the model is deployed online in the real world, the incoming data stream must be cached for a full sliding window before being fed into the model. This results in a 10x increase in memory consumption.
>
> **Q1.2: Why we can break the limitations of local window?**
>
> As two reviewers have question about this question, we provide a consistent response in **Q2** of the "global" response. Thanks.
>
> **Q2: Necessity of the predicted trend component.**
>
> The core of this question is the same as **Q1**. Obtaining the trend component directly using the moving average, as done in the data preprocessing phase, is more direct. However, the method necessitates a sliding window length several times larger than the period. Otherwise, the "standard trend component" merely represents local smoothing of the original series, lacking genuine trend extraction. In the case of long-period time series, employing such a large sliding window presents many problems.
>
> **Q3: Necessity of dynamically modifying information bottleneck and the  principle of hyperparameters setting.**
>
> Below, we shall expound the reasonableness of Challenge 2 (Line 34 in the paper). Firstly, we express regret for confusion arising from our imprecise statement (line 52 in the paper). The term "inference" would be more appropriately substituted with "revision". In truth, the requirement to adjust information bottlenecks not only exists but is also common in real world:
>
> - Training: when deploying the anomaly detection model on a novel scenario, guaranteeing the availability of the initial information bottleneck size proves challenging. Consequently, it becomes necessary to make multiple hyperparameter adjustments to revise the model.
> - Inference: in a real industry environment, temporal patterns are likely to change over time. The information bottleneck size set during training may not always yield satisfactory performance during inference. Hence, it becomes necessary to adjust it to revise the model.
>
> Unlike previous methods that require retraining the model when adjusting the information bottleneck, our approach can flexibly modify the information bottleneck without the need for retraining. Consequently, our method substantially reduces the training cost associated with information bottleneck modifications, even approaching close to zero cost.
>
> Regarding the principles of setting hyperparameters, a thorough analysis and discussion have already provided in Appendix F.
>
> **Q4: Verify the superiority of the model in reducing training cost.**
>
> Our algorithm inherently possesses the capacity to reduce the high training cost associated with adjusting the information bottleneck. This superiority stems from the fundamental design logic of our method.
>
> Specifically, in contrast to previous methods, our approach can flexibly adjust the bottleneck without the need for retraining. In other words, our method incurs close to zero cost solely for adjusting information bottleneck. Furthermore, as shown in Figure 4(b) in the paper, our algorithm exhibits superior performance and greater insensitivity to parameters. This advantage is evident as our algorithm achieves excellent performance with fewer adjustments. The negligible cost of a single adjustment and the limited number of adjustments required both substantiate the superiority of our design.
>
> **Q5: Insufficient investigation of previous research.**
>
> Apologies for not being able to provide a comprehensive comparison of previous research. In the revised version, we will include relevant analyses about TFAD [3].
>
> TFAD notes the significance of trend in anomaly detection. However, for series decomposition, they employ the method based on HP filter during both training and inference. This static method lacks applicability for real-world scenarios where data is updated in real-time. Furthermore, they use the distance between the context window and suspect window as the anomaly score. This method heavily relies on the assumption that the context window must be normal, which is challenging to ensure in complex real world.
>
> We conduct comparison experiments between our method and TFAD using the official open-source code. The experimental results are summarized in the table below.  The detailed version is available in the "global" response (Table 1 in the attached PDF). As analyzed in the previous paragraph, our method achieves the best experimental results.
>
> | Method |  PSM (F1)  |  SMD (F1)  | SWaT (F1)  | Average (F1) |
> | :----: | :--------: | :--------: | :--------: | :----------: |
> |  TFAD  |   0.7520   |   0.7149   |   0.6953   |    0.7207    |
> |  Ours  | **0.7609** | **0.8682** | **0.7812** |  **0.8034**  |
>
> **References:**
>
> [1] Hang Zhao, et al. “Multivariate time-series anomaly detection via graph attention network.”, In *ICDM*, 2020.
>
> [2] Chaoli Zhang, et al. “TFAD: A decomposition time series anomaly detection architecture with time-frequency analysis.”, In *CIKM*, 2022.

---

### Author Rebuttal · Authors · 2023-08-06

**Q1: Experiments on computational cost.**

As real-world time series datasets can be large-scale and complex, we supplement the measures of training time, inference time, and model size for deep learning-based models on the SMD dataset. The experimental results are presented in the subsequent table.

Statistically, the training time for these models remains below 10 minutes, which is acceptable for real-world deployment and maintenance. The utilization of attention and its variants in the backbone network of our model leads to its larger size compared to previous algorithms. Thanks to the highly parallelized nature of the attention mechanism, both the training time and inference time of our model remain competitive. Additionally, it is worth mentioning that there has been substantial recent work [1] on transformer linearization, which may aid in reducing the burden of our model. We plan to explore this aspect in our future research.

|       Method        | Training Time (s) | Inference Time (s) | Model Size (MB) |
| :-----------------: | :---------------: | :----------------: | :-------------: |
|         VAE         |      157.91       |       30.90        |      0.02       |
|      DeepSVDD       |     **60.70**     |     **12.61**      |    **0.01**     |
|       LSTM-AE       |      283.61       |       72.82        |      0.04       |
|      MTAD-GAT       |      188.52       |       60.23        |      1.20       |
| Anomaly Transformer |      422.43       |       94.36        |      28.15      |
|        TFAD         |      315.39       |       38.54        |      1.04       |
|        Ours         |      399.32       |       104.12       |     109.35      |

**Q2: Details of dynamic decomposition.**

The moving average method is utilized **solely in the data preprocessing stage during training** to extract the trend and stable components. At this stage, the training data has not undergone slicing based on sliding windows, thus avoiding any limitations of local sliding window. The specific decomposition algorithm employed at this stage is not a concern, as our primary objective is to generate a labeled stable component for model training. We can also use other more advanced decomposition algorithms, such as STL [2] or HP filter [3], to obtain more precise label.

With a labeled stable component, our model can start training. Essentially, the core of the dynamic decomposition module is **learning a mapping function from timestamps to the stable component**. During training, the model learns this mapping function iteratively. During testing, it can **directly map the current input series to its stable component with this function**. It is worth noting that we no longer require the labeled stable component during testing. With the help of mapping rather than moving averages within localized windows, our model allows for precise decomposition of time series within smaller sliding window (or even single point).

**References:**

[1] Haixu Wu, et al. “Flowformer: Linearizing transformers with conservation flows.” In *ICML*, 2022.

[2] Cleveland Robert, et al. “STL: A seasonal-trend decomposition procedure based on loess.” In *Journal of official statistics*, 1990.

[3] Robert J Hodrick, et al. “Postwar US business cycles: an empirical investigation.” In *Journal of Money, credit, and Banking*, 1997.

---

### Decision · Program_Chairs · 2023-09-21

**Decision:**

Accept (poster)

**Comment:**

The paper proposes the Decomposition with Diffusion Reconstruction (D3R) for anomaly detection in multivariate time series data. They decompose the time series into the trend and stable components, to capture long periods. They then use a reconstruction component involving a diffusion model based on stacked spatio-temporal transformer blocks. Experiments show convincing improvements over a good range of baselines on real world anomaly datasets.

There reviews contain 5 positive reviews (77766) and 1 negative review (3). On the positive side, reviewers noted that the method is well-motivated, novel, well-written, and shows good performance. On the negative side, the reviewer raises a number of weaknesses, including motivation (the challenge / significance of long periods, trends, and dynamically modifying the information bottleneck being debatable) and comparison to previous research (both in terms of experiments and discussion). In the authors' response, they give some specific examples to justify the motivation, and provide additional results showing computation time (comparable to other transformer-based approaches), and provide an additional baseline, TFAD. Overall, in my view the weaknesses have been reasonably addressed, and considering the strengths of the paper and near consensus between reviewers / AC, I recommend accepting the paper.

Authors are requested to note the suggested improvements / action items arising from discussion with the reviewers, such as adding the TFAD discussion / analyses and discussion of long periods to the final manuscript, and other improvements (e.g. minor writing fixes) arising from reviewer discussion.